# Comment on Cuesta et al. An Assessment of a New Rapid Multiplex PCR Assay for the Diagnosis of Meningoencephalitis. *Diagnostics* 2024, *14*, 802

**DOI:** 10.3390/diagnostics14171884

**Published:** 2024-08-28

**Authors:** Glaucia Paranhos-Baccalà, Tania Curião, Julien Textoris, Florence Allantaz

**Affiliations:** Global and EME Medical Affairs Departments, bioMérieux SA, 100 Rue Louis Pasteur, 69280 Marcy l’Etoile, France; glaucia.baccala@biomerieux.com (G.P.-B.); tania.curiao@biomerieux.com (T.C.); julien.textoris@biomerieux.com (J.T.)

In April 2024, the manuscript by Cuesta et al. was published in the Diagnostics (MDPI) journal [1], assessing the performance of the QIAstat-Dx^®^ Meningitis/Encephalitis (ME) Panel compared to the BIOFIRE^®^ FILMARRAY^®^ Meningitis/Encephalitis (ME) Panel and conventional methods. Fifty CSF samples from patients with suspected central nervous system (CNS) infections were studied. The sensitivities and specificities were 85.19% (CI 95%, 55.9–90.2) and 57.14% (CI 95%, 29.6–70.3), respectively, for the BIOFIRE^®^ FILMARRAY^®^ Meningitis/Encephalitis (ME) Panel. Discrepancies between both panels, including high numbers of false positive results for HSV-1 using the BIOFIRE^®^ ME Panel were also described in the study. As a company dedicated to ensuring the safety of our products for patients, we would like to address some concerns regarding the study design and reported results, as they do not align with the existing evidence.

The BIOFIRE^®^ ME Panel has 94.2% sensitivity and 99.8% specificity [2]. To date, there have been two systematic reviews on the clinical accuracy of the BIOFIRE^®^ ME Panel [3,4]. Trujillo-Gomez et al. performed a meta-analysis on diagnostic test accuracy including 19 studies, representing 11,351 participants [3]. They reported a combined sensitivity of more than 89% and specificity of more than 97% compared to two different reference tests. The diagnostic accuracy review from Tansarli et al. pooled eight studies (3059 patients) and showed that both the sensitivity and specificity of the BIOFIRE^®^ ME panel were >90% [4]. These two reviews analyzed the clinical accuracy of the ME Panel in studies comprising 14,410 patients with CNS infections, demonstrating significantly higher BIOFIRE^®^ ME Panel performances compared to the results reported by Cuesta et al. [1].

Furthermore, three manuscripts [5,6,7] have been published showing comparable results in terms of performance between the QIAstat^®^-Dx ME Panel and the BIOFIRE^®^ ME Panel. In Cuesta et al. [1], only 50 samples were analyzed, and the samples tested using the QIAstat^®^-Dx ME panel were selected based on availability. The samples underwent different testing times and intermediate freezing, potentially affecting the quality and results, as shown by Gaensbauer et al. in a study conducted on fresh CSF samples from 1387 adults and pediatric patients [8]. The QIAstat^®^ ME Panel testing was more controlled when compared to the BIOFIRE^®^ ME Panel, which was run in real-time. Additionally, the high percentage of positive samples studied was not representative of the typical clinical laboratory conditions, which might have skewed the results. The patient population was unusual, with many CSF samples showing normal or minimally altered biochemistry and a high percentage of immunosuppressed patients, increasing the risk of false positives for HSV-1 [9]. The concomitant HSV-1 detection in cases of other documented causes of infection might be due to the re-activation of HSV-1 [10] and potential traumatic lumbar puncture.

The discordant HSV-1 results described by Cuesta et al. [1] contradict Sundelin et al. [5], where both QIAstat^®^ and BIOFIRE^®^ ME panels showed a similar performance, with 20/585 clinical samples positive for HSV-1 by both. Trujilo-Gomez et al.’s meta-analysis [3], encompassing data from 6883 patients across three studies, reported HSV-1 sensitivities over 80% in two studies and over 60% in the third, with specificity exceeding 95% in all. Tansarli et al. [4] found similar results, pointing to very good specificity but a suboptimal sensitivity for HSV-1, which does not align with Cuesta et al.’s findings [1]. Furthermore, it is not clear how the clinical adjudication was performed on the samples and if it was conducted without the results of a different testing.

Cuesta et al. [1] claimed that the results obtained on the QIAstat^®^-Dx ME panel were accompanied by an amplification curve and its corresponding Ct value, which allow for a better microbiological interpretation together with other clinical data. This option is also available on BIOFIRE^®^ systems, using a recently launched application (BIOFIRE^®^ FIREWORKS^TM^) that allows the user to visualize the amplification curves and Cp values for positive qualitative results. It should be noted, however, that several studies have failed to demonstrate a clear correlation between viral load in CSF and outcome in patients with herpetic encephalitis [11,12,13,14,15,16,17,18]. 

In Cuesta et al., several samples had polymicrobial detections, which is very intriguing and mostly uncommon in meningitis/encephalitis cases as previously reported [3,4,19]. Contamination issues during the workflow cannot be discarded. Further investigations and reruns of these samples with discordant results could have helped in elucidating the polymicrobial results; unfortunately, this was not described in the paper. 

Syndromic testing has transformed the diagnosis of CNS infections, significantly reducing the time to effective therapy. Robust evidence links syndromic testing with decreased length of hospital stays and reduced antimicrobial use in both adults and children with meningitis/encephalitis [19,20,21]. However, clinical correlation remains essential; diagnosis should not rely solely on laboratory results. Implementing syndromic tests within a diagnostic stewardship framework ensures that the right test is used for the right patient at the right time, optimizing result interpretation, clinical impact, and cost-effectiveness. This should be paired with an antimicrobial stewardship program for timely antimicrobial adjustments based on diagnostic interpretations.

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
