# Peer review of "Comment on Cuesta et al. An Assessment of a New Rapid Multiplex PCR Assay for the Diagnosis of Meningoencephalitis. Diagnostics 2024, 14, 802"

_diagnostics, 2024, doi:10.3390/diagnostics14171884_

Round 1

Reviewer 1 Report

Comments and Suggestions for Authors

In this letter to the Editor representatives of the company, who produced the assays used in a study by the authors explained in a detailed and evidence based way appropriately previous data from meta-analyses. They raised the very important point of Implementing syndromic tests i.e. using test systems like the one their company produced in the correct clinical patient context and with documentation of adherence to appropriate procedures.

Response from the authors of the original publication is relevant. It should be published alongside the company's letter.

Both author groups should in addition consider that the positive HSV1 result in a documented other cause of encephalitis may not be a false positive result but in fact represent concomitant re-activation of HSV 1 within the central nervous system as explained previously:

Eisenhut M. Mycoplasma pneumoniae encephalitis and reactivation of herpes simplex virus. Pediatrics. 2007 Jun;119(6):1256-7

Author Response

In this letter to the Editor representatives of the company, who produced the assays used in a study by the authors explained in a detailed and evidence based way appropriately previous data from meta-analyses. They raised the very important point of Implementing syndromic tests i.e. using test systems like the one their company produced in the correct clinical patient context and with documentation of adherence to appropriate procedures.

Response from the authors of the original publication is relevant. It should be published alongside the company's letter.

Both author groups should in addition consider that the positive HSV1 result in a documented other cause of encephalitis may not be a false positive result but in fact represent concomitant re-activation of HSV 1 within the central nervous system as explained previously:

Eisenhut M. Mycoplasma pneumoniae encephalitis and reactivation of herpes simplex virus. Pediatrics. 2007 Jun;119(6):1256-7

Response: 

Thank you for your comments. We believe it is important to highlight that it is essential to use these types of diagnostic tests within a diagnostic stewardship framework, i. e. in a judicious approach, only under high suspicion of infection and as a tool complementing other clinical and laboratory data.

Thank you for the suggestion of considering the possibility of re-activation of HSV-1 under other causes of CNS infection. Following your comment, a sentence considering the HSV-1 re-activation and also having traumatic blood CSF samples was added in the revised version of the reply to the authors on line 42. 

Reviewer 2 Report

Comments and Suggestions for Authors

The criticisms to the paper from Cuesta et al are well documented and supported by papers included i the reference list. I think that Cuesta et al have to respond to the criticisms as concern the sensitivity of the involved panel, the lack of the amplification curve and Ct value application to the Biofire, the fact that Biofire has been run in real-time and the fact that samples are unusually polymicrobic.

Author Response

The criticisms to the paper from Cuesta et al are well documented and supported by papers included i the reference list. I think that Cuesta et al have to respond to the criticisms as concern the sensitivity of the involved panel, the lack of the amplification curve and Ct value application to the Biofire, the fact that Biofire has been run in real-time and the fact that samples are unusually polymicrobic.

Response: Thank you for your feedback. We agree that a reply from the original authors to the points raised would be helpful to clarify the scientific community about the false flaws in the ME Panel and ensure a credible and pragmatic approach to diagnostic testing to benefit patients as much as possible.